# HSP70 via HIF-1 α SUMOylation inhibits ferroptosis inducing lung cancer recurrence after insufficient radiofrequency ablation

**Bin Peng, Xiean Ling, Tonghai Huang\*, Jun Wan** \*

Department of Thoracic Surgery, Shenzhen People's Hospital, Shenzhen, China

\* tonghaihuang@aliyun.com (TH); wanjun6226@126.com (JW)

**Data Availability Statement:** The original data set generated and/or analyzed during the current study can be obtained in the SRA repository, login

## Abstract

Radiofrequency ablation (RFA) is an effective and feasible therapy for lung cancer, but accelerated progression of residual non-small cell lung cancer (NSCLC) after incomplete radiofrequency ablation (RFA) has frequently been reported. A previous study reported that HSP70 and HIF-1α were highly expressed in areas with incomplete RFA. Therefore, we sought to elucidate the regulatory effect of the HIF-1α/HSP70 pathway on lung cancer recurrence after incomplete radiofrequency ablation. In this study, we found that knockdown of HSP70 can reduce sumo 1, sumo 2/3 (marker of SUMOylation) of HIF-1α and inhibit A549 cell proliferation and migration under heat stress conditions (used to simulate incomplete RFA in vitro). We observed that knockdown of HSP70 altered the expression of ferroptosis-related proteins and genes (SLC7A11 and ACSL3), and the RNA-seq results showed that knockdown of HSP70 activated the ferroptosis pathway, further confirming that HSP70 regulates ferroptosis. In summary, HSP70, via HIF-1α SUMOylation, inhibited ferroptosis, inducing lung cancer recurrence after radiofrequency ablation. The study reveals a new direction for further research on therapeutic targets to suppress lung cancer recurrence and provides a theoretical foundation for further clinical studies.

## Introduction

Lung cancer remains the primary cause of cancer-related deaths worldwide. Lung cancer is separated into two broad histologic types: non-small cell lung cancer (NSCLC) and small cell lung cancer (SCLC). The most common type is NSCLC [1]. Radiofrequency ablation (RFA) is an important replacement for surgery in patients with lung cancer. The National Comprehensive Cancer Network guidelines indicate that RFA can be a therapeutic option for patients with a poor physical status [2]. Although RFA plays an important role in lung cancer therapy, it is associated with a high recurrence rate [3]. A previous study reported accelerated progression of residual non-small cell lung cancer after RFA [4].

HIF-1α is an important factor regulated in hypoxic environments. Under RFA therapy, lung cancer cells have been shown to transform into a proliferative heat-tolerant subtype and result in high HIF-1α expression [4]. A previous study reported that HIF-1α increases angiogenesis, activates procancer signalling pathways, and mediates the activation of transcription

number [PRJNA943582], the link is http://www.ncbi.nlm.nih.gov/bioproject/943582.

**Funding:** The present study was supported by the Special Fundamental Research Project of Shenzhen Science and Technology Plan (Natural Science Foundation) No. JCYJ20190807144201675. This is the sources of support received during this study. The funder provided support in the form of salaries for author [JW], but did not have any additional role in the study design, data collection and analysis, decision to publish, or preparation of the manuscript. The specific roles of these authors are articulated in the 'author contributions' section. There was no additional external funding received for this study.

**Competing interests:** The present study was supported by the Special Fundamental Research Project of Shenzhen Science and Technology Plan (Natural Science Foundation) No. JCYJ20190807144201675.

factors, thereby promoting tumour progression [5]. HIF-1α interferes with immunosuppressive effects by regulating the expression of programmed cell death-ligand 1 and CD73, resulting in anti-inflammatory effects on the tumour microenvironment [6]. The HIF-1α protein is degraded rapidly under normoxic conditions by the ubiquitin–proteasome system [7, 8]. Thus, the stability of HIF-1α is the primary factor determining the function of HIF-1. HSP90 is involved in the maintenance of HIF-1α stabilization, which is mediated by ubiquitination and proteasomal degradation. HSP90 binds to the PAS domain of HIF-1α and stabilizes the HIF-1α subunit primarily under normoxic conditions [9]. Specifically, high HIF-1α expression has been found in many subtypes of lung cancer [10].

Ferroptosis, a form of regulated necrotic cell death, is an iron-dependent and reactive oxygen species (ROS)-related mode of cell death characterized by a decrease in mitochondrial cristae, rupture of the outer mitochondrial membrane, and condensation of mitochondrial membranes [11]. Depletion of cysteine/GSH leads to decreased GPX4 activation and specifically induces this form of cell death [12]. The function of GPX4, an enzyme that catalyses the reduction of lipid peroxides at the expense of the oxidation of two molecules of GSH and inhibits the activation of arachidonic acid (AA) metabolic enzymes, is a key factor in ferroptosis. If lipid peroxides cannot be reduced effectively, they activate a cascade in which $Fe^{2+}$ generates ROS to oxidize lipids, finally inducing ferroptosis. Many diseases, such as cancer, diabetes and cardiovascular disease, have been reported to be associated with ferroptosis [13]. Ferroptosis is a new promising target for antitumour therapy. Interestingly, both activation and inhibition of ferroptosis can be utilized in anticancer therapies. On the one hand, ferroptotic damage can trigger inflammation-associated immunosuppression in the tumour microenvironment, thus contributing to tumour growth. On the other hand, some approved drugs can induce ferroptosis and suppress tumour growth [14]. The extent to which ferroptosis affects tumour biology is unclear, although several studies have reported important correlations between cancer-related genes and ferroptosis.

Our previous study found that incomplete RFA increased the proliferation of residual lung cancer cells via the HSP70/HIF-1α axis [15]. This study aims to further reveal the underlying molecular mechanism and correlations among local recurrence, HIF-1α and ferroptosis.

## Materials and methods

### Animals and cells

Male BALB/c nude Crlj mice were obtained from Beijing Vital River LaboratoryAnimal Technology Co., Ltd. (Beijing, China). Mice were maintained under pathogen-free conditions in accordance with established institutional guidance and approved protocols. All experiments were carried out using 3- to 4-week-old mice weighing 14–16 g. The animal study was reviewed and approved by Laboratory Animal Management and Experiment Committee of Guangdong Zhongke Yinghai Technology Co., Ltd (no.20230208-sr-mouser-1).

Animals were housed under standard laboratory conditions. Human non-small cell lung cancer A549 cells were purchased from Procell Life Science & Technology (Wuhan, China). The cells were cultured with RPMI-1640 medium containing 10% FBS (foetal bovine serum, Gibco, USA) at 37˚C in 5% $CO_2$.

### Establishment of a nude mouse model with human non-small cell lung cancer A549 cells

All animals were acclimated for at least 7 days prior to intravenous (i.v.) injection of A549 cells. On Day 1, A549 cells were collected and resuspended in phosphate-buffered saline (PBS; Thermo Fisher Scientific) at a concentration of $2 \times 10^7$ cells/ml. Cell suspensions (100 μl) were

administered to each mouse via tail vein injection. Tumour metastasis was monitored by in vivo imaging at 1 or 3 weeks following the injection.

## Surgical procedures and RFA treatment

All operations were performed under anaesthesia, and mice were intraperitoneally injected with a ketamine/methylthiazole mixed anaesthetic. During the experiment, all animals received humane care according to the guidelines of animal research institutions.

RFA treatment was applied according to the method of a previous study [16]. RFA was performed on the left lung using a bipolar RFA device (Blade Opto-Electronic Technology Development Co., China) with a 10 mm microprobe. To simulate the clinical settings of tumour recurrence following RFA, we performed RFA for 180 sec at a power of 1 W for tumour ablation.

## H&E staining

Lung tissue was fixed in 10% neutral formalin solution and embedded in paraffin, and the paraffin sections were cut into 4 μm slices. The slices were dewaxed with xylene and washed with an ethanol gradient prior to staining with haematoxylin and eosin (Thermo, USA). The cells were fixed with neutral resin and observed under a microscope (Nikon, Japan).

## Western blot

The cells were washed in ice-cold PBS and boiled in loading buffer, and proteins were then separated by 8% SDS–PAGE, after which they were transferred onto polyvinylidene difluoride membranes (IPVH00010, Millipore, Germany). The membranes were incubated overnight with the appropriate primary antibodies prior to incubation with alkaline phosphatase-conjugated secondary antibodies. The intensities of the bands were quantified using the National Institutes of Health (NIH) ImageJ program. Antibody information is shown in S1 Table.

## qPCR

The mouse tissue samples and cell samples were lysed with TRIzol (Invitrogen, USA) for collection of total RNA according to the manufacturer's instructions. The cells and tissue samples were rinsed with phosphate-buffered saline (PBS) twice, TRIzol was added to the lysed cells (the tissue samples were ground first), and total RNA was extracted following the phenol/chloroform method. cDNA was synthesized using the PrimeScript™ RT Reagent Kit (Takara, Japan). Real-time quantitative PCR was performed on the StepOne™ Real-Time PCR System (Life Technologies, USA). Each cDNA sample was analysed in triplicate, and target mRNA expression was normalized to GAPDH expression. The mRNA expression level in each sample was calculated by using the $2^{-\Delta\Delta Ct}$ method. Primer sequences are shown in S2 Table.

## RNA-seq

Total RNA was extracted by TRIzol. Library construction, sequencing and analysis were performed by Chi-Biotech (Shenzhen, China). Briefly, the libraries were sequenced using the Illumina platform. Differentially expressed genes (DEGs) were screened based on the following criteria: $p < 0.05$ and absolute fold change $>2.0$. The size factors of every RNA-seq dataset were determined using the R package Limma for differential expression analysis, and the package clusterProfiler was then used for enrichment analysis. The intracellular signalling pathways were identified using Kyoto Encyclopedia of Genes and Genomes (KEGG) analysis based on the DEGs. Gene Ontology (GO) term analysis, KEGG pathway analysis and gene set enrichment analysis (GSEA) were performed by TopGO and clusterProfiler package.

## Invasion assay

The invasion assay was performed using Transwell plates (Corning Costar, USA). In brief, cells were seeded into chambers coated with Matrigel in complete medium supplemented with 10% FBS. The cells were cultured for 48 h. Cells that did not penetrate the filter were removed by wiping, and cells on the lower surface of the filter were stained with 0.4% crystal violet. The invaded cells were counted under a microscope from 5 fields in a single chamber.

## CCK8 assay

Cells were seeded in a 96-well plate (100 μL/well). The plates were preincubated in an incubator (37˚C, 5% $CO_2$). Then, 10 μL of CCK solution was added to each well (carefully to avoid generating bubbles, which affect the results). The culture plates were incubated in an incubator at 37˚C and 5% $CO_2$ for 2 hours. A microplate reader was used to measure the absorbance at 450 nm.

## ROS assay

Cells (90% confluent) were collected, washed with warm DPBS and harvested by trypsinization. The cells were pelleted in 5 ml polystyrene tubes by centrifugation for 5 min at $200 \times g$ and room temperature, and the supernatant was discarded. The cell pellet was washed with 4 ml DPBS and centrifuged for 5 min at $200 \times g$ and room temperature. Propidium iodide (final concentration = 1 μg/ml) was added to the pellet and placed on ice (in the dark). The propidium iodide staining procedure was performed immediately before analysis on a flow cytometer. Cell viability was assessed by propidium iodide staining. PI fluorescence was assessed in a FACSVantage SE instrument (Becton Dickinson). Viable cells, which are PI-negative, were selected by FACS gating. In these viable cells, we recorded the dichlorodihydrofluorescein fluorescence.

## Co-IP

The IP lysis buffer was supplemented with a proteasome inhibitor as required by the kit. The monoclonal antibody against the marker protein was added quantitatively. The samples were incubated with end-over-end rotation at 4˚C. The protein-antibody complexes were transferred to a centrifuge tube and incubated with protein A/G agarose in a sealed container with rotation for 1 h. Elution agarose was centrifuged at $800 \times g$ for 3 min at 4˚C. Then, Western blot analysis was performed after boiling the immunoprecipitation products. Interactions between proteins were detected by incubation with an anti-rat SUMO1 monoclonal antibody (1:800) and anti-rat SUMO2/3 polyclonal antibody (1:600).

## Statistical analysis

One-way analysis of variance (ANOVA) was used for statistical analysis, and T-test was used for comparison between groups. Data are expressed as mean ± standard deviation. Differences were considered statistically significant at $P < 0.05$.

## Results

### Activation of HIF-1α signalling and inhibition of ferroptosis after insufficient RFA of lung cancer

Notably, the morphological characteristics of lung cells varied among normal tissue, untreated lung cancer cells and lung cancer cells after insufficient RFA (Fig 1A). Compared with the normal group, the lung cancer group showed tightly arranged tumour cells with enlarged nuclei

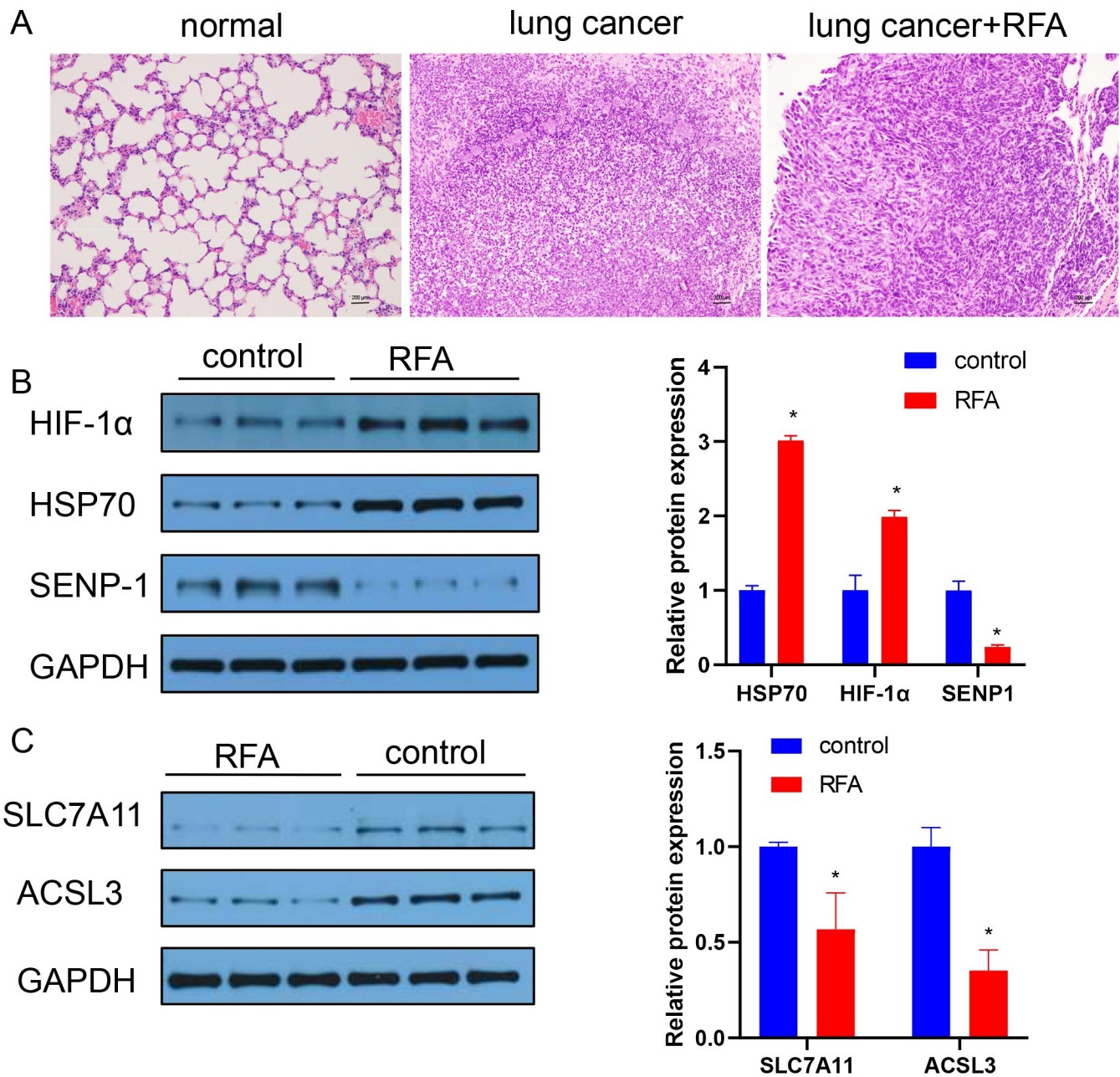

**Fig 1. Changes in Lung Cancer Cells after insufficient RFA of lung cancer.** (A) HE staining was used for histomorphological observation of lung tissue. Normal: normal mouse lung tissue; lung cancer: mouse lung cancer tissue; lung cancer+ RFA: mouse lung cancer tissue after RFA treatment. (B) Western blot analysis of HIF-1α, HSP70, and SENP-1 expression in mice. (C) Western blot analysis of SLC7A11 and ACSL3 expression in mice. control: mice with lung cancer; RFA: mice with lung cancer treated with RFA. "*" $p<0.05$; "**" $p<0.01$.

and an increased nuclear–cytoplasmic ratio. After insufficient RFA, the cells were not arranged as densely.

Additionally, we found that the expression levels of HIF-1α and its binding protein HSP70 were increased after insufficient RFA in the lung cancer mouse model (Fig 1B). The expression of SENP1, a protein promoting HIF-1α ubiquitination-mediated degradation, was decreased after RFA treatment (Fig 1B). The expression of some proteins of the ferroptosis pathway

(SLC7A11 and ACSL3) was also decreased after insufficient RFA of lung cancer cells (Fig 1C). These results suggest that insufficient RFA can activate HIF-1α signalling and inhibit ferroptosis.

## HSP70 knockdown inhibits the proliferation and invasion of A549 cells after heat treatment via HIF-1α SUMOylation

Then, we explored the effect of HSP70 knockdown on heat-treated A549 lung cancer cell lines. We cultured cells at 47°C to simulate insufficient RFA. HSP70 siRNA was transfected into A549 cells to inhibit HIF-1α expression (our previous study demonstrated that HSP70 promotes the expression of HIF-1α [17]). Compared with those in the control group, heat treatment increased the expression levels of HIF-1α and HSP70 protein and mRNA, and HSP70 knockdown treatment reversed this effect (Fig 2A and 2B). The invasion assay results showed that heat treatment decreased the cell invasion ability, possibly due to cell death and decreased viability under heat stress (Fig 2C). The invasion ability was further decreased after HSP70 inhibition (Fig 2C). Additionally, when we assessed the proliferation of A549 cells, we found that heat treatment inhibited cell proliferation. HSP70 knockdown significantly reduced proliferation compared with that in the heat-treated group (Fig 2D). In brief, heat treatment decreased cell viability, and promoted the expression of HIF-1α and HSP70. Knockdown of HSP70 revealed that heat treatment decreased the expression of HIF-1α and HSP70, enhancing the inhibitory effect of heat treatment on A549 cells.

In addition, we also observed that many immune chemokine genes were upregulated after heat treatment and downregulated after inhibiting HSP70 (S1 Fig). Our previous study proved that HSP70 promotes HIF-1α expression via SUMOylation [17]. Similarly, in this study, we found that heat stress increased the SUMO level of HIF-1α (Fig 2E). The SUMO level of HIF-1α was decreased after knockdown of HSP70 (Fig 2E). Some SUMO-associated proteins, such as SENP1, Ubc9 and RanBP2, also showed a tendency to return to normal levels after knockdown of HSP70 (Fig 2F). Therefore, we concluded that HSP70 knockdown inhibits the proliferation and invasion of A549 cells after heat treatment by reducing HIF-1α SUMOylation.

## Inhibition of HSP70 activates the ferroptosis pathway

HIF-1 signalling modulates many downstream genes, including many genes involved in iron metabolism and ferroptosis, such as SLC40A1 and FPN1 [18–20]. In our study, some ferroptosis-associated genes were also differentially expressed after combined heat treatment and siRNA-HSP70 transfection (Fig 3A). The expression levels of two proteins that inhibit ferroptosis, SLC7A11 and ACSL3, were decreased in the heat+siHSP70 group compared with the heat-treated group (Fig 3B). The heat+siHSP70 group showed higher ROS levels than the heat-treated group (Fig 3C). We found that the heat-treated group had higher ALC7A11 and ACSL3 expression levels and ROS levels than the control group (Fig 3B, 3C), possibly due to apoptosis and cell death caused by heat stress and ferroptosis triggered by heat stress. These results indicated that HSP70 has an important impact on the ferroptosis pathway.

## HSP70 knockdown changed the gene expression profile of A549 cells after heat treatment

To further explore the changes caused by heat treatment and inhibition of HSP70, we performed mRNA-seq to assess changes in mRNA expression, pathways and biological functions. Hundreds of DEGs were identified after heat treatment and knockdown of HSP70, with only a very small overlap among the three analysed groups (Fig 4A). We identified 146 upregulated

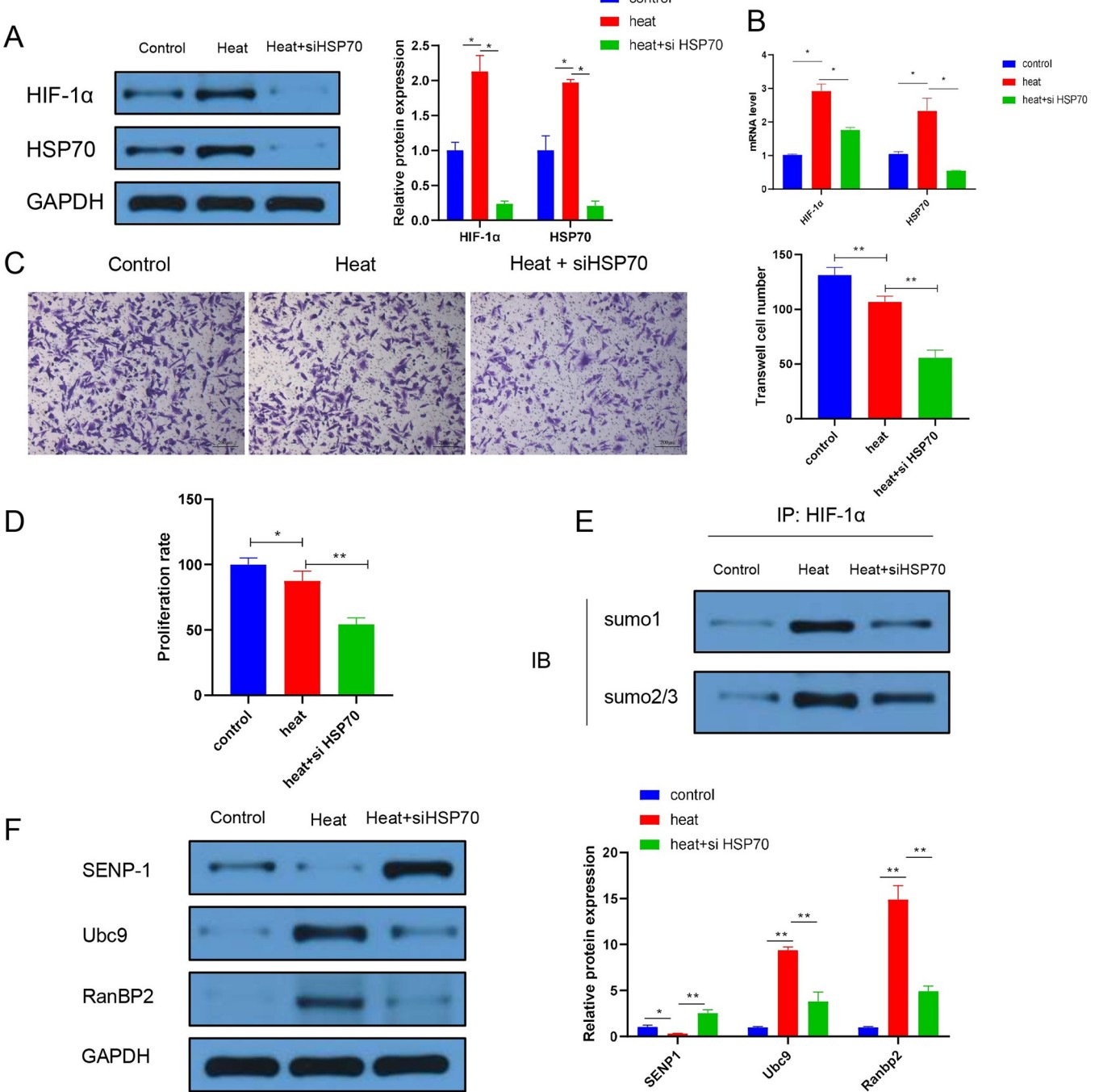

**Fig 2. Effect of HSP70 knockdown on heat-treated A549 lung cancer cell lines.** (A) Western blot analysis of HIF-1α and HSP70 expression in the three analysed groups. (B) qPCR analysis of HIF-1α and HSP70 mRNA expression. (C) Transwell invasion assay of the A549 cells in the three groups (200x magnification). (D) CCK8 proliferation assay of A549 cells in the three groups. (E) Co-IP of HIF-1α with sumo1 or sumo2/3 using anti-HIF-1α antibody-conjugated beads (IP) followed by immunoblot analysis (IB) with anti-sumo1 or anti-sumo2/3 antibodies. (F) Western blot analysis of SENP-1, Ubc9, and RanBP2 expression in the three groups. Control: A549 cells with no treatment; Heat: A549 cells with thermal challenge; Heat+siHSP70: A549 cells transfected with siRNA HSP70 after thermal challenge. "*" p<0.05; "**" p<0.01.

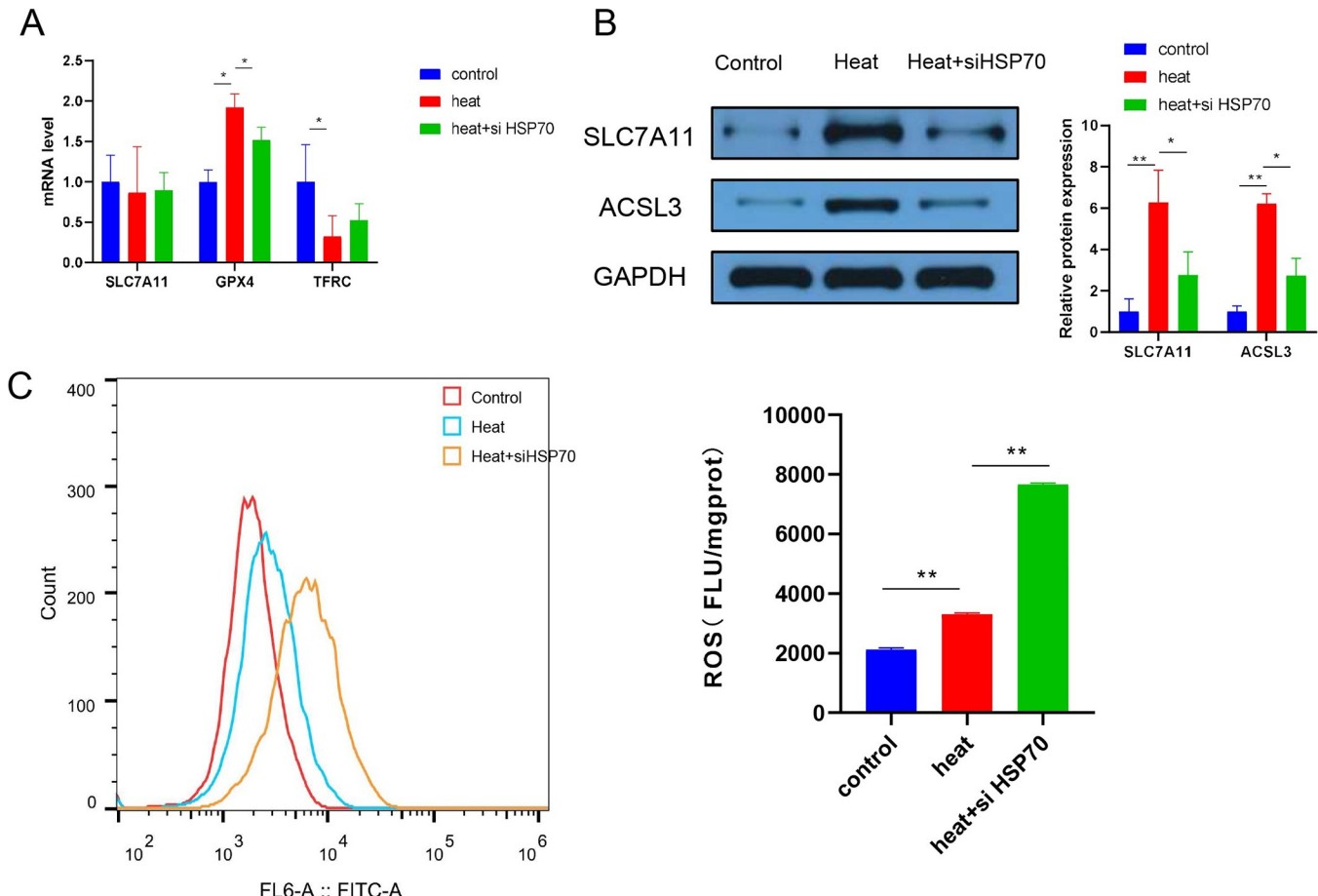

**Fig 3. Effect of HSP70 inhibition on iron death response in A549 lung cancer cell lines.** (A) qPCR analysis of SLC7A11, GPX4, and TFRC mRNA expression. (B) Western blot analysis of SLC7A11 and ACSL3 expression in the three groups. (C) ROS detection by flow cytometry. Control: A549 cells with no treatment; Heat: A549 cells with thermal challenge; Heat+siHSP70: A549 cells transfected with siRNA HSP70 after thermal challenge. "*" p<0.05; "**" p<0.01.

genes and 68 downregulated genes after knockdown of HSP70 (Fig 4B). To explore the functional categories of the DEGs among those three groups, GO and KEGG analyses were performed. A total of 169 KEGG pathways were enriched in the heat + siHSP70 group, including the ferroptosis pathway and IL-17 pathway (Fig 4C). The DEGs involved in these KEGG pathways might undergo alterations in expression and play a vital role in A549 cells after heat treatment and knockdown of HSP70. To avoid overlooking genes with subtle changes in expression but an important and profound influence, GSEA was performed to identify pathway changes. We found that the ferroptosis pathway was activated after inhibition of HSP70 (Fig 4D). Some differentially expressed genes have been reported to be associated with ferroptosis. We measured the mRNA expression levels of 10 genes that inhibit the activation of ferritin decarboxylase by PCR. These genes were upregulated in the heat+siHSP70 group, consistent with the results of RNA-seq (Fig 4E). These results further indicate that HSP70 can induce the recurrence of lung cancer after RFA by inhibiting ferroptosis.

## Discussion

RFA is a safe and effective approach for lung cancer therapy. However, rapid growth of residual lung tumours after RFA has been observed in many clinical centres, while the detailed

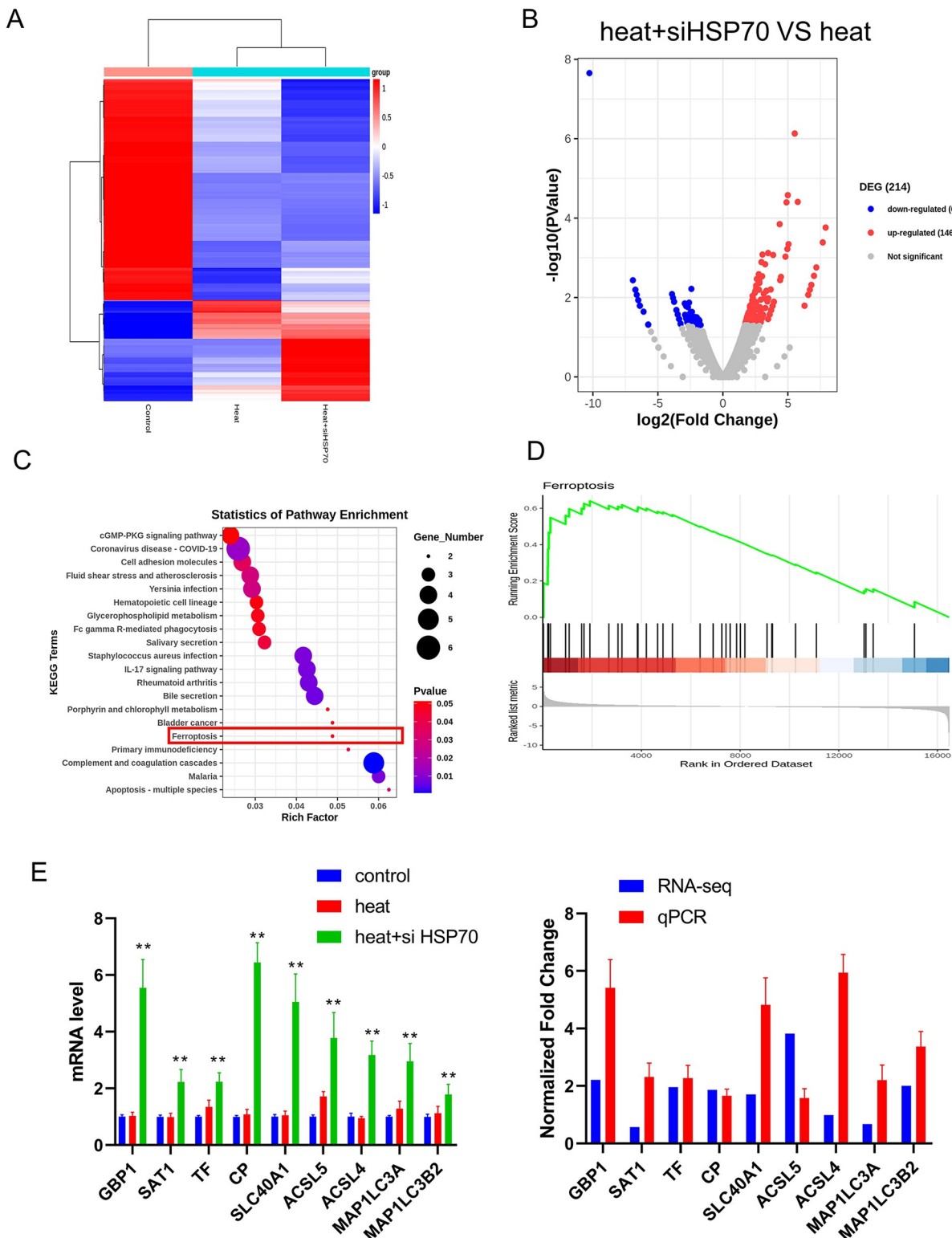

**Fig 4. HSP70 knockdown changed the gene expression profile of A549 cells after heat treatment.** (A) Heatmap of the DEGs among the three groups. (B) Volcano plots presenting the differences between the heat and heat+siHSP70 groups. (C) KEGG analysis of the DEGs between the heat and heat+siHSP70 groups. (E) GSEA of the Heat+ siHSP70 vs. Heat groups. (F) qPCR verification of the expression of 10 genes from the mRNA-seq data. Left: qPCR analysis of the three groups; Right: Normalized fold changes in gene expression between the heat-treated group and the heat group; Control: A549 cells with no treatment; Heat: A549 cells with thermal challenge; Heat+siHSP70: A549 cells transfected with siRNA HSP70 after thermal challenge. "*" p<0.05; "**" p<0.01.

mechanisms have not been well illustrated. Currently, it has been reported that the high expression of HIF-1α remains a major cause of recurrence after radiofrequency ablation for lung cancer [4]. HIF-1α, a subunit of HIF-1, can promote the malignant progression of various cancers by facilitating cell proliferation, metastasis, and tumour angiogenesis [21]. Insufficient RFA has been reported to trigger angiogenesis and proliferation in residual hepatocellular carcinoma via upregulation of HIF-1α [22, 23]. Our present study further revealed that insufficient RFA (or heat stress) increased the expression of HIF-1α, as found in other studies [4, 24]. These results confirmed that the HIF-1α pathway is directly involved in the regulation of lung cancer recurrence after RFA, but the specific mechanism is still unclear. Elucidating the regulatory effect of the HIF-1α pathway on lung cancer recurrence after radiofrequency ablation is of great importance for improving the prognosis of NSCLC patients.

HSP70, a heat shock response protein that plays an important role in posttranscriptional regulation, has been observed to modulate HIF-1α expression [15, 25]. We knockdownHSP70 in A549 cells after heat treatment and observed that the mRNA and protein levels of HIF-1α were significantly decreased and that cell proliferation and invasion were decreased. A study showed that small ubiquitin-related modifier (SUMO) can posttranslationally modify HIF-1α to improve the stability of HIF-1α and increase the transcriptional activity and expression level of HIF-1α [26]. Our Co-IP results showed that heat treatment increased the SUMOylation of HIF-1α and that the SUMO-associated enzymes SEPN1, Ubc9 and RanBP2 were differentially expressed in response to heat stress. However, in the heat+si HSP70 group, Ubc9 expression, RanBP2 expression and SUMOylation of HIF-1α decreased. This result indicated that HSP70 knockdown inhibited the proliferation and invasion of A549 cells after heat treatment via HIF-1α SUMOylation.

Ferroptosis is a unique form of iron-dependent oxidative cell death that plays an important role in the development of a variety of cancers [27]. Studies have suggested that suppression of ferroptosis promotes tumour invasion and metastasis [28, 29]. HIF-1α regulates iron metabolism-related gene expression through hypoxia response elements (HREs) and promotes tumorigenesis by inhibiting ferroptosis [30, 31]. In recent years, an increasing number of studies have shown that HSPs participate in the pathophysiological process of ferroptosis [32]. They play different roles in the occurrence, development and regulation of ferroptosis. In our study, both the WB and RNA-seq results showed that the ferroptosis pathway was activated after knockdown of HSP70. The main mechanism of ferroptosis is as follows: unsaturated fatty acids highly abundant on the cell membrane induce lipid peroxidation under the action of divalent iron or ester oxygenase [33]. SLC7A11 and ACSL3 (two factors involved in lipid oxidation and ferroptosis) were downregulated. These results showed that inhibition of HSP70 activates the ferroptosis pathway. In addition, we performed mRNA-seq to assess changes in mRNA expression, pathways and biological functions in HSP70 knockdown A549 cells after heat treatment. The results further indicated that HSP70 can induce the recurrence of lung cancer after insufficient RFA by inhibiting ferroptosis.

In summary, insufficient RFA or heat stress can induce high HSP70/HIF-1α expression, and knocking down HSP70 can decrease HIF-1α SUMOylation and activates ferroptosis. This result indicates that HSP70, via HIF-1α SUMOylation, inhibits ferroptosis, inducing lung cancer recurrence after insufficient radiofrequency ablation. However, the specific mechanism requires further study. The study reveals a new direction for further research on therapeutic targets to suppress lung cancer recurrence and provides a theoretical foundation for further clinical studies.

## Supporting information

**S1 Fig. qPCR analysis of immune chemokine genes expression after heat treatment.**
(TIF)

**S1 Table. Antibody information.**
(DOCX)

**S2 Table. Primer information.**
(DOCX)

**S1 File. Original blot or gel images.**
(PDF)

## Acknowledgments

Thanks for every patient participated in this study. It is their disease data that lays the foundation for the subsequent treatment of patients.

## Author Contributions

**Conceptualization:** Xiean Ling, Jun Wan.

**Data curation:** Jun Wan.

**Formal analysis:** Bin Peng, Xiean Ling.

**Funding acquisition:** Jun Wan.

**Investigation:** Bin Peng.

**Project administration:** Bin Peng, Xiean Ling.

**Resources:** Tonghai Huang.

**Software:** Xiean Ling.

**Supervision:** Xiean Ling.

**Visualization:** Tonghai Huang, Jun Wan.

**Writing – original draft:** Xiean Ling.

**Writing – review & editing:** Tonghai Huang, Jun Wan.

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
