## [Decision Letter · Decision Letter 0]

23 Aug 2023

PONE-D-23-06260HSP70 via HIF-1 α SUMOylation inhibits ferroptosis inducing lung cancer recurrence after insufficient radiofrequency ablationPLOS ONE

Dear Dr. Wan,

Thank you for submitting your manuscript to PLOS ONE. After careful consideration, we feel that it has merit but does not fully meet PLOS ONE’s publication criteria as it currently stands. Therefore, we invite you to submit a revised version of the manuscript that addresses the points raised during the review process.

We look forward to receiving your revised manuscript.

Kind regards,

A R M Ruhul Amin, Ph.D.

Academic Editor

PLOS ONE

Journal Requirements:

https://www.researchgate.net/publication/359872340_Novel_Therapeutic_Savior_for_Osteosarcoma_The_Endorsement_of_Ferroptosis

https://academic.oup.com/abbs/article/48/4/371/1754676?login=false

https://www.sciencedirect.com/science/article/pii/S2213231722000842?via%3Dihub

In your revision ensure you cite all your sources (including your own works), and quote or rephrase any duplicated text outside the methods section. Further consideration is dependent on these concerns being addressed.

“This study was supported by Science Foundation of Shenzhen (No. JCYJ20180302144713444).”

6. PLOS requires an ORCID iD for the corresponding author in Editorial Manager on papers submitted after December 6th, 2016. Please ensure that you have an ORCID iD and that it is validated in Editorial Manager. To do this, go to ‘Update my Information’ (in the upper left-hand corner of the main menu), and click on the Fetch/Validate link next to the ORCID field. This will take you to the ORCID site and allow you to create a new iD or authenticate a pre-existing iD in Editorial Manager. Please see the following video for instructions on linking an ORCID iD to your Editorial Manager account: https://www.youtube.com/watch?v=_xcclfuvtxQ.

7. Your ethics statement should only appear in the Methods section of your manuscript. If your ethics statement is written in any section besides the Methods, please move it to the Methods section and delete it from any other section. Please ensure that your ethics statement is included in your manuscript, as the ethics statement entered into the online submission form will not be published alongside your manuscript.

8. PLOS ONE now requires that authors provide the original uncropped and unadjusted images underlying all blot or gel results reported in a submission’s figures or Supporting Information files. This policy and the journal’s other requirements for blot/gel reporting and figure preparation are described in detail at https://journals.plos.org/plosone/s/figures#loc-blot-and-gel-reporting-requirements and https://journals.plos.org/plosone/s/figures#loc-preparing-figures-from-image-files. When you submit your revised manuscript, please ensure that your figures adhere fully to these guidelines and provide the original underlying images for all blot or gel data reported in your submission. See the following link for instructions on providing the original image data: https://journals.plos.org/plosone/s/figures#loc-original-images-for-blots-and-gels.

10. Thank you for stating the following in the Competing Interests section:

“The present study was supported by the Special Fundamental Research Project of Shenzhen Science and Technology Plan (Natural Science Foundation) No. JCYJ20190807144201675.”

We note that one or more of the authors have an affiliation to the commercial funders of this research study : Natural Science Foundation

Reviewers' comments:

Reviewer's Responses to Questions

**Comments to the Author**

1. Is the manuscript technically sound, and do the data support the conclusions?

Reviewer #1: Partly

2. Has the statistical analysis been performed appropriately and rigorously? 

Reviewer #1: I Don't Know

3. Have the authors made all data underlying the findings in their manuscript fully available?

Reviewer #1: Yes

4. Is the manuscript presented in an intelligible fashion and written in standard English?

Reviewer #1: Yes

5. Review Comments to the Author

Reviewer #1: In this study, the authors used cell heat stress model to imitate incomplete radiofrequency ablation (RFA) conditions. They found that insufficient RFA (heat stress) may up-regulate the expression levels of HSP70 and HIF-1��in A549 cells.They further showed that inhibition of HSP70 decreased HIF-1� SUMOylation and suppressed ferroptosis of A549 cells, as indicated by decreased expressions of SENP1(SUMO-associated), SLC7A11, ACSL3 (ferroptosis pathway genes). The results of this study are interesting and the manuscript is quite well written. However, I have a couple of concerns / comments.

1. The results of this study are uncertain, because all data are based mainly on only single type cell (A549 cells). The authors need to conduct similar experiments using another type of NSCLC cell line.

2. It is also difficult to obtain conclusion based on current data, because results are basically obtained from HSP70 knockdown experiments. Whether overexpression of HSP70 will demonstrate opposite results is unclear.

3. Abstract: the names of SUMU-associated and ferroptosis-related markers should be mentioned.

4. Materials and Methods: Statistical methods should be mentioned

5. Discussion: in the second paragraph, “posttranslational modification”, should be “posttranscriptional regulation”

6. Figure legends: It would be better to give each Fig. a title. For example: Fig.1 Heat stress up-regulated HSP70 and HIF-1� expression but inhibited the expression of SENP-1, SCL7A11, and ACSL3…..

6. PLOS authors have the option to publish the peer review history of their article (what does this mean?). If published, this will include your full peer review and any attached files.

Reviewer #1: **Yes: **Rihong Zhai

---

## [Author Response · Author response to Decision Letter 0]

17 Oct 2023

See document "1 Response to Reviewers.docx" for details.

Question and response: 

Response: We have revised the manuscript to fit the style of PLOS ONE.

Response: Information needed for editor is listed in Materials and Methods-Animals and Cells, Surgical procedures and RFA treatment.

Response: We have compared the manuscript with several papers presented by the editors, and have corrected those with a high degree of repetition.

Response: We added this to the Funding and cover letter.

Response: The original data set generated and/or analyzed during the current study can be obtained in the SRA repository, login number [PRJNA943582], the link is http://www.ncbi.nlm.nih.gov/bioproject/943582.

6. PLOS requires an ORCID iD for the corresponding author in Editorial Manager on papers submitted after December 6th, 2016. Please ensure that you have an ORCID iD and that it is validated in Editorial Manager. 

Response: We have already set it up.

7. Your ethics statement should only appear in the Methods section of your manuscript. 

Response: We have modified it.

8. PLOS ONE now requires that authors provide the original uncropped and unadjusted images underlying all blot or gel results reported in a submission’s figures or Supporting Information files. 

Response: Our original uncropped and unadjusted blot or gel image data are placed in the Supporting information file.

9. Please include captions for your Supporting Information files at the end of your manuscript, and update any in-text citations to match accordingly. 

Response: We have added the title of the supporting information document at the end of the manuscript.

10. Thank you for stating the following in the Competing Interests section:

“The present study was supported by the Special Fundamental Research Project of Shenzhen Science and Technology Plan (Natural Science Foundation) No. JCYJ20190807144201675.”

We note that one or more of the authors have an affiliation to the commercial funders of this research study: Natural Science Foundation

Response: We have already made a statement about this in the Funding section.

Reviewers' comments:

Reviewer #1:

1. The results of this study are uncertain, because all data are based mainly on only single type cell (A549 cells). The authors need to conduct similar experiments using another type of NSCLC cell line.

Response: Many thanks to reviewers for your suggestions! It has been reported in the literature that after radiofrequency ablation, patients have elevated levels of serum HSP70, a good biomarker for lung cancer [1]. Our previous study on human NSCLC NCI-H1650 cell line found that HSP70/HIF-1α induced accelerated proliferation and angiogenic potential in residual lung cancer after RFA treatment, and its expression was regulated by PI3K/Akt signaling pathway [2]. Studies on A549-H, CCL-185-H and H358-H cells found that insufficient RFA could promote NSCLC growth in vitro and in vivo by upregulating HIF-1α via PI3K / Akt signaling [3]. And our present study using the A549 cell line similarly found the regulatory role of the HIF-1α/HSP70 pathway on the recurrence of lung cancer after incomplete radiofrequency ablation. 

We believe that these studies have demonstrated the regulatory mechanism of the HIF-1α/HSP70 pathway in the recurrence of lung cancer in different cell types after incomplete radiofrequency ablation.

Nevertheless, we are again grateful for the suggestions made by the reviewers, which will remind us to place more value on the rigor of the protocol and logical completeness in our future studies.

Reference: 

[1] Haen SP, Gouttefangeas C, Schmidt D, et al. Elevated serum levels of heat shock protein 70 can be detected after radiofrequency ablation. Cell Stress Chaperones. 2011;16(5):495-504. doi:10.1007/s12192-011-0261-y

[2] Wan J, Wu W, Huang Y, Ge W, Liu S. Incomplete radiofrequency ablation accelerates proliferation and angiogenesis of residual lung carcinomas via HSP70/HIF-1α. Oncol Rep. 2016;36(2):659-668. doi:10.3892/or.2016.4858

[3] Wan J, Wu W, Chen Y, Kang N, Zhang R. Insufficient radiofrequency ablation promotes the growth of non-small cell lung cancer cells through PI3K/Akt/HIF-1α signals. Acta Biochim Biophys Sin (Shanghai). 2016;48(4):371-377. doi:10.1093/abbs/gmw005

2. It is also difficult to obtain conclusion based on current data, because results are basically obtained from HSP70 knockdown experiments. Whether overexpression of HSP70 will demonstrate opposite results is unclear.

Response: Thanks for your valuable comments. In the cellular experimental design, we detected the expression of HSP70 in different treatments (Control, Heat, Heat+siHSP70) by Western blot assay (Figure 2A, figure below). The results showed that HSP70 was significantly up-regulated in the Heat treatment group compared to the control group, suggesting that this treatment had already indicated the overexpression treatment of HSP70. Therefore, we did not conduct further experiments on overexpression of HSP70 separately. We believe that the results of the study have demonstrated the role of HSP70.

3. Abstract: the names of SUMU-associated and ferroptosis-related markers should be mentioned.

Response: Thank you for your suggestions, which we have supplemented SUMO-associated and ferroptosis-related markers in the Abstract section

4. Materials and Methods: Statistical methods should be mentioned

Response: Thanks for pointing this out, we have added statistical methods in the Materials & Methods section!

5. Discussion: in the second paragraph, “posttranslational modification”, should be “posttranscriptional regulation”

Response: We have made the changes. 

6. Figure legends: It would be better to give each Fig. a title. For example: Fig.1 Heat stress up-regulated HSP70 and HIF-1� expression but inhibited the expression of SENP-1, SCL7A11, and ACSL3…..

Response: Thank you for your suggestions. We have updated the figure legends of all the figures.

The above are our responses to editorial and reviewer comments. We look forward to receiving your further decisions.

---

## [Decision Letter · Decision Letter 1]

30 Oct 2023

HSP70 via HIF-1 α SUMOylation inhibits ferroptosis inducing lung cancer recurrence after insufficient radiofrequency ablation

PONE-D-23-06260R1

Dear Dr. Wan,

We’re pleased to inform you that your manuscript has been judged scientifically suitable for publication and will be formally accepted for publication once it meets all outstanding technical requirements.

Kind regards,

A R M Ruhul Amin, Ph.D.

Academic Editor

PLOS ONE

Additional Editor Comments (optional):

Reviewers' comments:

Reviewer's Responses to Questions

**Comments to the Author**

1. If the authors have adequately addressed your comments raised in a previous round of review and you feel that this manuscript is now acceptable for publication, you may indicate that here to bypass the “Comments to the Author” section, enter your conflict of interest statement in the “Confidential to Editor” section, and submit your "Accept" recommendation.

Reviewer #1: All comments have been addressed

2. Is the manuscript technically sound, and do the data support the conclusions?

Reviewer #1: Yes

3. Has the statistical analysis been performed appropriately and rigorously? 

Reviewer #1: Yes

4. Have the authors made all data underlying the findings in their manuscript fully available?

Reviewer #1: Yes

5. Is the manuscript presented in an intelligible fashion and written in standard English?

Reviewer #1: Yes

6. Review Comments to the Author

Reviewer #1: The authors have addressed all the comments on the manuscirpt and the quality of the manuscript has been improved after revision.

7. PLOS authors have the option to publish the peer review history of their article (what does this mean?). If published, this will include your full peer review and any attached files.

Reviewer #1: **Yes: **Rihong Zhai

---

## [Editor Report · Acceptance letter]

2 Nov 2023

PONE-D-23-06260R1 

HSP70 via HIF-1 α SUMOylation inhibits ferroptosis inducing lung cancer recurrence after insufficient radiofrequency ablation 

Dear Dr. Wan:

I'm pleased to inform you that your manuscript has been deemed suitable for publication in PLOS ONE. Congratulations! Your manuscript is now with our production department. 

Kind regards, 

on behalf of

Dr A R M Ruhul Amin 

Academic Editor

PLOS ONE